# Antibacterial Potential of Microwave-Assisted Extraction Prepared Hydrolates from Different *Salvia* Species

**DOI:** 10.3390/plants12061325

**Published:** 2023-03-15

**Authors:** Eva Ürgeová, Ľubica Uváčková, Miroslava Vaneková, Tibor Maliar

**Affiliations:** Faculty of Natural Sciences, University of SS. Cyril and Methodius, J. Herdu 2, SK-917 01 Trnava, Slovakia; lubica.uvackova@ucm.sk (Ľ.U.); tibor.maliar@ucm.sk (T.M.)

**Keywords:** *Salvia* hydrolates, antimicrobial activity, antioxidant activity, bacteria

## Abstract

Salvia is a widely used herb that also contains essential oils and other valuable compounds. In this work, the hydrolates of five *Salvia* sp. were evaluated for their potential antimicrobial and antioxidant activity against four bacterial strains. The hydrolates were obtained from fresh leaves by microwave-assisted extraction. Chemical composition analysis by gas chromatography and mass spectrometry revealed that their major constituents were isopulegol (38.2–57.1%), 1,8-cineole (4.7–19.6%), and thujone (5.6–14.1%). The minimum inhibitory concentration (MIC) of the plant hydrolates was tested by the microdilution method at concentrations ranging from 1.0 to 512 μg/mL. The hydrolates prepared from *Salvia officinalis* and *S. sclarea* showed inhibitory activity on the tested Gram-positive and Gram-negative bacteria, taxon *Salvia nemorosa* showed inhibitory activity only partially. The hydrolate of *S. divinorum* had practically no antibacterial effect. *Enterobacter asburiae* was the only bacterium for which we found sensitivity to the hydrolate of *S. aethiopis*, with a MIC50 value of 216.59 µL/mL. The antioxidant activity of the hydrolates was low, ranging from 6.4 to 23.3%. Therefore, salvia hydrolates could be used as antimicrobial agents in medicine, cosmetics, and food preservation.

## 1. Introduction

In natural medicine, plants and their products are used to solve many health problems. Nowadays, few people know about natural medicine, and even fewer actively use it to help with health problems. It is important to choose appropriate herbal products for a particular health problem, and the choice depends on the presence and amount of biologically active components responsible for antimicrobial, antiviral, antiparasitic, anticarcinogenic, and antioxidant activity [1]. We are encountering great interest in these natural products, mainly due to the resistance of microorganisms to antibiotics. In addition, there is a growing interest in producing safer food crops and developing new antibacterial agents that can be used in food production.

The genus *Salvia*, which belongs to the *Lamiaceae* family, includes over 1000 species distributed worldwide [2,3,4]. Some members of the genus are used in cosmetics, as flavourings, or as medicines. The members of this genus produce many useful secondary metabolites—the essential oils. The composition of essential oils varies depending on the species and variety [5]. They may contain about 20–60 constituents in completely different concentrations, with usually 2–3 major constituents in relatively high concentrations. Essential oils contain constituents such as thujone, camphor, 1,8-cineole, and borneol in *Salvia officinalis* [6,7]; β-caryophyllene, germacrene D, and caryophylene oxide in *S. nemorosa* [8]; α-copaene and β-cubenene δ-cadinene in *S. aethiopis* [9]; linalool, linalyl acetate, α-terpineol, and sclareol in *S. sclarea* [10]; psychoactive diterpenes salvinorin A and salvinorin B in *S. divinorum* [11], but also carnosic acid, nicotinic acid, rosmarinic and oleanic acids, carnazole, tannins, saponins, bitter compounds, and polyphenols [12].

Salvia essential oils and extracts are effective against a variety of organisms, including bacteria, fungi, viruses, and insects. Salvia is characterized by antibacterial, antifungal, antiviral, and anti-inflammatory properties [13,14,15,16]. In recent decades, scientific research has focused its interest on sage essential oils as natural sources of antimicrobial compounds. Many studies have described the inhibitory effect of sage essential oils on Gram-positive bacteria (*Microccous luteus*, *Staphylococcus aureus*, *Bacilus cereus*, *Bacilus subtilis*, *Clostridium perfingens*, *Streptococcus pneumoniae*, and *Mycobacterium smegmatis*) [2,17,18,19,20,21] and Gram-negative bacteria (*Klepsiella oxytoca* and *Aeromonas hydrophyla*) [2]. Much less attention is paid to the byproduct of essential oil production—hydrolates, which contain many bioactive hydrophilic substances [22]. A hydrolate (H) is internationally defined as the distilled aromatic water that remains after hydro- or steam distillation and separation of the essential oil (EO) (ISO 9235:2013). Hydrolates form a heterogeneous suspension of essential oil and water-soluble substances obtained via distillation and exhibit interesting antimicrobial activities against microorganisms [23,24,25]. Hydrolates have an intense aroma, and it is known that the difference in the composition of a hydrolate from that of essential oils is mainly quantitative [26,27]. Considering their aroma and hydrophilicity, hydrolates offer promising prospects for food processing applications and are capable of controlling *Listeria* biofilms and, in particular, preventing their formation [28]. Hydrolates are easy and inexpensive to produce and appear to be less toxic to human health compared to essential oils [25]. This suggests that they can be used in medicine as potential antimicrobial agents. In this study, we aimed to determine the antimicrobial activity of hydrolates prepared by microwave-assisted extraction from different *Salvia* species against selected strains of Gram-positive bacteria (*Micrococcus luteus* DSM 1790 and *Bacillus subtilis* DSM 5552) and strains of Gram-negative bacteria (*Escherichia coli* CCM 3954 and *Enterobacter asburiae* CCM 8546).

## 2. Results

### 2.1. Chemical Composition

The chemical composition of hydrolates was determined by gas chromatography followed by mass spectrometry (GC/MS). Table 1 shows the results of the relative composition of hydrolates obtained by microwave-assisted extraction of fresh plant material. In total, from 20 different possible compounds, 13 were identified. Most compounds (eleven) were identified in the hydrolates of *S. divinorum* and *S. sclarea*, and at least six were identified in the hydrolate of *S. officinalis*. All hydrolates primarily contain the isoprenoid isopulegol (38.22–57.11%) (Figure 1a), followed by 1,8-cineole (Figure 1b) in *S. officinalis* (19. 57%) and *S. nemorosa* (15.67%) hydrolates, and thujone (Figure 1c), the main component present in hydrolates of *S. nemorosa*, *S. aethiopis*, and *S. divinorum* in amounts greater than 10%.

### 2.2. Antimicrobial Activity

The minimum inhibitory concentrations (MIC) MIC50 and MIC90 (expressed in μL/mL) against four bacterial strains are summarized in Table 2. The MIC tests showed the strongest antibacterial activity of *S. officinalis* hydrolate on *M. luteus* with MIC50 (5.69 µL/mL) and MIC90 (7.81 µL/mL) values.

Based on the measured absorption, we focused on the percentage expression of bacterial growth in the presence of hydrolates. We determined 100% bacterial growth in pure MHB and statistically compared this value with bacterial growth when treated with antimicrobials (the hydrolates). Growth of all strains was inhibited by hydrolates of *S. officinalis* (Figure 2) and *S. sclarea* (Figure 3).

The results showed that hydrolates of *S. nemorosa* inhibited more Gram-positive than Gram-negative bacterial strains (Figure 4).

The least inhibitory effect was observed for the strain *E. asburiae* when treated with *S. nemorosa* hydrolate. The growth of *M. luteus* was completely inhibited by *S. officinalis* and by *S. nemorosa* hydrolates at a concentration of 125 µL/mL.

### 2.3. Antioxidant Activity

In the analysis of *Salvia* hydrolates, the DPPH method was used. The hydrolates showed varying antioxidant activity (Table 3).

The results showed the strongest antioxidant activity of *S. officinalis* hydrolate at 23.25 ± 0.26% of inhibition.

## 3. Discussion

Salvia, an aromatic and medicinal herb, is known for many purposes [29]. The preparations of sage are known for their antibacterial activity against various bacteria [30,31]. Hydrolates have recently attracted attention for their antimicrobial activity, especially against pathogenic and spoilage microorganisms, including bacteria and fungi [32]. The chemotype of a hydrolate is important to understand the mechanisms of its biological activity. For example, phenolic compounds are considered potent antibacterial agents that tend to attack the outer membrane of bacteria. They also have effects on the structural and functional properties of the cytoplasmic membrane [25]. Other hydrolates have shown antibacterial effects as they accumulate in the cell membrane, affecting its integrity and causing the release of intracellular material [33]. It is known that the volatile profiles of hydrolates depend on the origin of the plant material [32].

By evaluating the chemical composition analysis via GC/MS, the relative proportion of the different compounds with potential antimicrobial activity was determined. All hydrolates primarily contain isopulegol and the presence of other isoprenoids. More than 10% of 1,8-cineole contains hydrolates of *S. officinalis* (19.57%), *S. nemorosa* (15.67%), and *S. aethiopis* (11.75%). Thujone, the main component of *Salvia* essential oils, was present in hydrolates of *S. nemorosa*, *S. aethiopis*, *S. sclarea*, and *S. divinorum* in amounts greater than 10%. Other scientists also studied different salvia hydrolates. The results of [4] showed that the main constituents of *S. sclarea* and *S. officinalis* are 1,8-cineole, α-thujone, linalool, and borneol. The aromatic water obtained from a sage sample collected in Turkey showed camphor, 1,8-cineole, thujone, and borneol as the main constituents [34]. The results of 1,8-cineole and thujone were in agreement with our results. In contrast with [25], our results did not confirm the presence of carvacrol in hydrolates.

The ability of hydrolates to inhibit bacterial growth depends on their chemical composition and the bacterial strain. The antimicrobial activity of hydrolates is explained by secondary major and minor components, which, in some cases, coincide with those of essential oils (although in different amounts), while in others they are completely different [25]. According to the results of phytochemical analysis of five *Salvia* sp. hydrolates, the main components are terpenes and their derivatives—isopulegol, 1,8-cineole, thujone, borneol, and linalool. Antimicrobial activities have been reported for borneol and 1,8-cineole. The minor constituents such as 1,8-cineole might be involved in the antibacterial activity of hydrolates [25]. The antimicrobial activity of thujone has also been confirmed [35,36,37]. Some researchers suggested that the hydroxyl groups of eugenol could bind to proteins by preventing the action of microbial enzymes [38]. In this study, we tested hydrolates of five *Salvia* sp. We found no antimicrobial effect of the hydrolate of *S. divinorum* in this study where MIC50 and MIC90 were more than 500 µL/mL for three bacteria—*E. coli*, *M. luteus,* and *B. subtilis,* and a very low antimicrobial effect on *E. asburiae* with a MIC50 value of 325.61 µL/mL. Similarly, this strain was sensitive to *S. aethiopis* hydrolate, with a MIC50 value of 216.59 µL/mL. The Gram-positive strains were resistant to *S. aethiopis* hydrolates. All strains proved sensitive to hydrolates of *S. officinalis* with MIC50 value of 5.69–56.65 µL/mL and *S. sclarea* with MIC50 value of 74.99–141.41 µL/mL. *M. luteus* proved to be the most sensitive to salvia hydrolate with MIC50 value (5.69 µL/mL), followed by *B. subtilis* (18.43 µL/mL), *E. coli* (27.50 µL/mL) and *E. asburiae* (56.65 µL/mL). The Gram-positive strains, with the exception of the hydrolate of *S. officinalis*, were also more sensitive to hydrolates of *S. nemorosa* than Gram-negative strains. *E. asburiae* was the only one in which we observed sensitivity to *S. aethiopis* and *S. divinorum* hydrolates.

When we focused on monitoring the percentage growth of bacteria treated with hydrolates, we confirmed and supplemented the results obtained by MIC. The hydrolates of *S. officinalis* and *S. nemorosa* were able, at the lowest concentrations, to partially inhibit the growth of Gram-positive strains throughout the cultivation. Similarly, the hydrolate of *S. sclarea* was marginally effective against bacterial strains at low concentrations. Ovidi et al. [4] tested essential oils of *S. sclarea* and defined them as bactericidal against the susceptible bacterial strains. Significant activity of these essential oils against *S. aureus* and *S. epidermidis* strains (MIC values of 10.0 and 5.0 mg/mL, respectively) was found [39]. *Salvia* species is among the plants effective against bacterial strains. Sagdiç and Ozcan [40] investigated the ability of *S. fruticosa* Mill. and *S. aucheri* L. to control common bacteria. Other authors tested the antibacterial activity of hydrolates of *S. officinalis* L. [41,42,43,44]. Tornuk et al. [41] indicated that the application of hydrolates showed a remarkable inhibitory effect on *S. aureus*. Sage was used to control the growth of *S. typhimurium*, *E. coli* O157:H7, and *L. monocytogenes*. In general, approximately three log reductions were obtained in the pathogens populations by using hydrosols [25]. Tornuk et al. [41] investigated the inhibitory effect of sage against *S. typhimurium* and *E. coli* O157:H7. The treatment procedures resulted in a significant reduction in the tested pathogenic bacteria. Ozturk et al. [42] studied the effect of plant hydrolates obtained from sage. Sage hydrolates eliminated *L. monocytogenes*. In contrast, according to [45], hydrolates from *S. officinalis* were ineffective against *E. coli* and *Pseudomonas aeruginosa*. In other studies [40,44], sage hydrolate was reported to be ineffective against fifteen bacteria in the in vitro test.

The antioxidant properties of plants are of interest because of their potential use as natural additives. There are few studies addressing the antioxidant properties of hydrolates. Their results show moderate to very low antioxidant capacity [37,46]. These results are consistent with the results of our study (see Table 3). In previous studies, the authors of [47] investigated the antioxidant activity of 55 Turkish *Salvia* species. In their study, the DPPH radical scavenging activity of *S. nemorosa* was reported to be 90.75% at 100 µg/mL. In contrast, Jeshvaghani et al. [48] reported 53% at 500 µg/mL. In other studies, the IC50 value of leaf extract of *S. sclarea* was 58.20 µg/mL [49] and 25 µg/mL [50]. Tepe et al. [51] reported the antioxidant activity of *S. sclarea* to be 23.4% at 50 µg/mL. These values of Turkish sage were higher than those of Slovak sage. According to [51], extracts of *S. aethiopis* did not show radical scavenging activity. Our results could not confirm this.

## 4. Materials and Methods

### 4.1. Isolation of Hydrolates

We used the leaves of *S. officinalis*, *S. nemorosa*, *S. aethiopis*, *S. sclarea*, and *S. divinorum* obtained from the Gene Bank of the Slovak Republic at the Research Institute of Plant Production in Piešťany. We added about 200 g of fresh sage leaves to the extraction container. The extraction took place in a microwave oven Bosch FFL023MS2 (Gerlingen, Germany) at 800 W for 8 min in EssenEx^®^ 100A Essential Oil Extraction Kit (Corvallis, OR, USA). We used ice to condense the steam and oil. After extraction, we separated essential oils and hydrolates. We stored the sage hydrolates (25 mL from each batch) in dark-coloured bottles at 4 °C until we tested their antimicrobial activity.

### 4.2. GC- MS Analysis of Salvia Hydrolates

The constituents were identified and the relative composition of the salvia hydrolates was determined by gas chromatography followed by mass spectrometry (GC/MS), as described by [52]. Prior to injection, hydrolates were extracted in hexane in a ratio of 1.5:1 and dried over anhydrous sodium sulphate according to [37]. Analyses were carried out using an Agilent 7890A GC coupled to an Agilent MSD5975C MS detector (Agilent Technologies, Palo Alto, CA, USA) with a HP-5MS column (30 m × 0.25 mm, 0.25 mm film thickness). The analytical conditions were as follows: injector temperature 250 °C and the oven temperature from 60 °C to 231 °C at 3 °C/min. Components were identified by matching their mass spectra and retention indices with those of authentic samples and/or NIST/Wiley spectra libraries and available literature data [53]. Relative proportions were calculated by dividing the individual peak area by the total area of all peaks. Only compounds with more than 1% were considered.

### 4.3. Free Radical Scavenging Assays

The antioxidant activity of the studied hydrolates was determined using the 2,2-diphenyl-1-picrylhydrazyl radical (DPPH, Sigma-Aldrich, Germany) according to [54], with a modification for the microplate form. Briefly, 50 μL of the tested samples were added to 100 μL of a 1.2% DPPH-ethanol solution. The solution was shaken and incubated for 30 min in the dark at room temperature. Absorbance was measured at 517 nm using the Opsys MRTM microplate reader from Dynex (Chantilly, VA, USA). Antioxidant activity was expressed as a percentage (%) of scavenging activity: DPPH scavenging activity (%) = [(A_c_ − A_s_)/(A_c_ − A_b_)] × 100, where A_c_ is the absorbance of the control (absorbance of the DPPH solution with distilled water), A_s_ is the absorbance of the sample, and A_b_ is the absorbance of the blank (ethanol). The analyses were performed in triplicate.

### 4.4. Bacterial Strains

Gram-negative strains Escherichia coli CCM 3954 and Enterobacter asburiae CCM 8546, and Gram-positive strains Micrococcus luteus DSM 1790 and Bacillus subtilis DSM 5552 obtained from the Czech Collection of Microorganisms (CCM; Brno, Czech Republic) and the German Collection of Microorganisms and Cell Cultures (DSMZ, Braunschweig, Germany) were used in this study to determine the antibacterial activity of the hydrolates. Broth cultures of the test strains were grown overnight in Mueller–Hinton broth medium (Hi-medium, India) (MHB) at 37 °C for 16 h.

### 4.5. Preparation of the Inoculum

The overnight cultures were diluted in culture medium and adjusted to a final concentration of 5 × 10^5^ CFU/mL. This was confirmed by colony counting according to CLSI guidelines [55]. The same procedure for preparing the inoculum was used for all experiments performed in this study.

### 4.6. Determination of Minimum Inhibitory Concentration

The determination of the minimal inhibitory concentration (MIC) was performed via serial microdilution in 96-well microtiter plates using Mueller–Hinton broth. Briefly, the hydrolates were diluted in MHB to obtain an initial concentration of 512 μg/mL. The twofold serially diluted concentrations of all hydrolates ranged from 512 to 0.25 μg/mL. The final bacterial concentration was adjusted to 5 × 10^5^ CFU/mL. The wells in the last column of all plates served as positive controls for measuring the optical density of all strains tested, and the wells in the first column were used as sterility controls for MHB. Microplates were incubated at 37 °C for 18–20 h. The lowest concentration of antimicrobial agent that prevented bacterial growth was defined as the minimum inhibitory concentration (MIC). Optical density was measured at 405 nm using an Opsys MRTM microplate reader from Dynex (Chantilly, VA, USA). The 96-well microplates were measured before and after the experiment (approximately 18 h). Results were expressed as the mean of three replicates in three independent experiments.

### 4.7. Statistical Analysis

Statistical analysis was performed by probit analysis using the Statgraphic program (Statpoint technologies, Warrenton, VA, USA) according to [56], with some modifications.

## 5. Conclusions

This study focused on the chemical analysis of hydrolates obtained via microwave-assisted extraction of five *Salvia* species and the evaluation of their antimicrobial and antioxidant potential. Salvia hydrolates are a promising source of various compounds with potential antimicrobial activity and plant antioxidants such as isopulegol, 1,8-cineole, thujone, borneol, and linalool. The results of this study indicate that the hydrolates have promising antimicrobial activity. The highest antimicrobial activity was observed for *S. officinalis* hydrolate. The best minimum inhibitory concentration (MIC) was found against Gram-positive bacteria *M. luteus* DSM 1790 by *S. officinalis* hydrolate with MIC90 7.81 μg/mL. The interesting antimicrobial activity of *S. sclarea* and *S. nemorosa* hydrolates was also described. Our results showed a low antioxidant capacity of the hydrolates. Our postulated aims of the research in this study were achieved: We wanted to remark on the differences between various Salvia varieties from the chemical composition and antibacterial and antioxidant activity points of view. These were proved and described. This study suggests that hydrolates could be used, for example, as natural antimicrobial agents or food preservatives, but further testing is needed.

## Figures and Tables

**Figure 1 plants-12-01325-f001:**
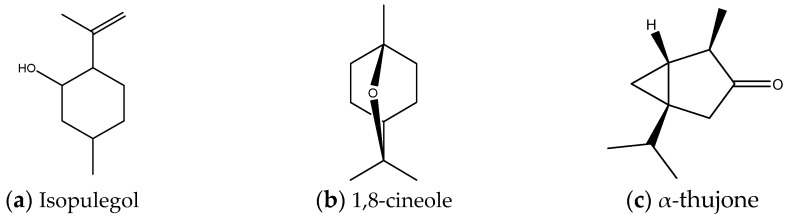
The main compounds in hydrolates.

**Figure 2 plants-12-01325-f002:**
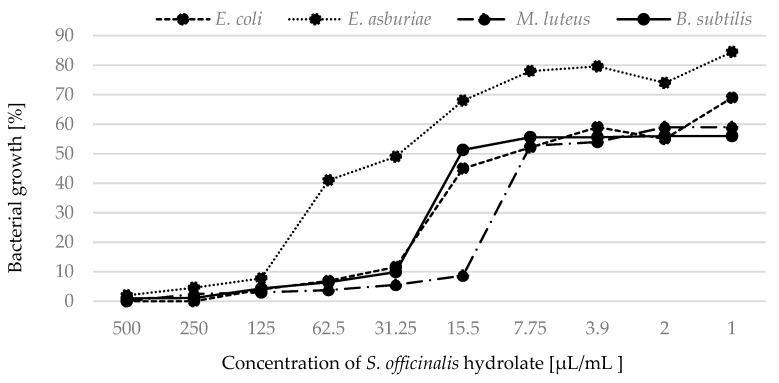
The growth of bacterial strains in the presence of *S. officinalis* hydrolate.

**Figure 3 plants-12-01325-f003:**
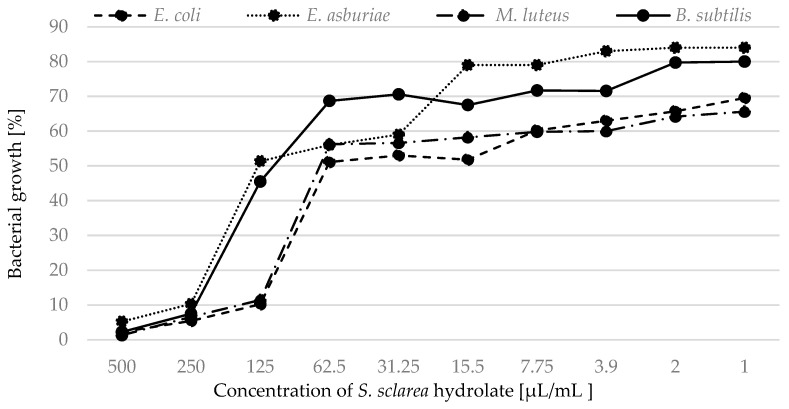
The growth of bacterial strains in the presence of *S. sclarea* hydrolate.

**Figure 4 plants-12-01325-f004:**
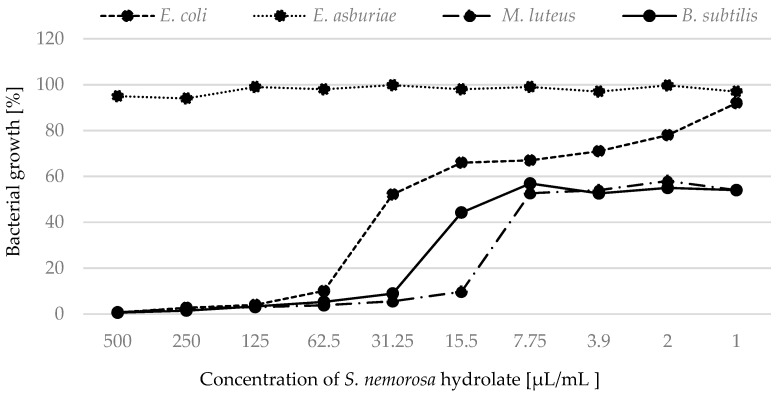
The growth of bacterial strains in the presence of *S. nemorosa* hydrolate.

**Table 1 plants-12-01325-t001:** Qualitative and quantitative analysis (%) of salvia hydrolates by GC/MS.

RI ^b^	Compound	*S. officinalis* ^c^	*S. nemorosa* ^c^	*S. aethiopis* ^c^	*S. sclarea* ^c^	*S. divinorum* ^c^
982	(-)-β-Pinene ^a^	-	1.31%	-	1.20%	3.15%
1034	1,8-Cineole ^a^	19.57%	15.67%	11.75%	4.71%	6.29%
1100	(-)-Linalool ^a^	4.01%	2.68%	3.70%	3.44%	3.40%
1117	α- and β-Thujone	5.57%	14.10%	13.78%	11.50%	10.95%
1137	Unidentified	-	-	1.03%	1.13%	1.21%
1146	(-)-Isopulegol	57.11%	47.20%	56.54%	40.26%	38.22%
1167	(-)-Borneol ^a^	5.41%	4.74%	5.06%	5.28%	6.55%
1178	Menthol (+/−) ^a^	-	2.02%	-	-	2.71%
1197	Carvomenthenol ^a^	1.12%	-	-	-	-
1403	Unidentified	-	-	-	-	1.38%
1413	Unidentified	-	1.50%	1.15%	-	-
1475	α-Cubebene	-	-	-	3.90%	4.10%
1517	Caryophyllene	-	-	-	3.28%	2.71%
1552	α-Caryophyllene	-	-	-	3.34%	3.04%
1579	Germacrene D	-	-	-	4.59%	4.21%
1585	Unidentified	-	-	-	-	1.29%
1623	Unidentified	-	-	-	-	1.10%
1625	Unidentified	-	-	-	-	1.20%
1680	Unidentified	-	-	-	1.28%	-
1689	Naphthalene	-	1.75%	1.26%	6.41%	-
-	Total	92.79%	90.95%	94.27%	90.31%	91.52%

Legend: ^a^ Identification confirmed by co-injection of the authentic standard; ^b^ RI: identification based on Kovat’s retention indices (HP-5MS capillary column) and mass spectra; ^c^ relative proportion was calculated in percentage by dividing the area of each peak by the total area of all peaks.

**Table 2 plants-12-01325-t002:** The minimum inhibitory concentration (MIC) of the tested hydrolates expressed in μL/mL.

Tested Hydrolates	*E. coli*		*E. asburiae*		*M. luteus*		*B. subtilis*	
	MIC50	MIC90	MIC50	MIC90	MIC50	MIC90	MIC50	MIC90
*S. officinalis*	27.50	42.12	56.65	123.16	5.69	7.81	18.43	31.25
*S. nemorosa*	38.82	62.58	>500	>500	8.51	15.63	11.26	28.14
*S. aethiopis*	>500	>500	216.59	407.13	>500	>500	>500	>500
*S. sclarea*	74.99	125.41	141.41	253.01	81.68	136.38	106.58	201.11
*S. divinorum*	>500	>500	325.61	472.61	>500	>500	>500	>500

**Table 3 plants-12-01325-t003:** Antioxidant activity expressed as percentage inhibition of DPPH radical at a concentration of 100 μg/mL of the tested hydrolates.

Tested Hydrolates	Antioxidant Activity (%)	TEAC (mg/mL)
*S. officinalis*	23.25 ± 0.26	52.16 ± 2.85
*S. nemorosa*	14.61 ± 0.09	31.08 ± 1.09
*S. aethiopis*	10.62 ± 0.96	24.12 ± 0.96
*S. sclarea*	18.99 ± 0.58	42.12 ± 1.58
*S. divinorum*	6.42 ± 0.57	14.41 ± 0.57

Legend: Values represent the average (standard deviations) for triplicate analyses. Each value is given as mean ± standard deviation.

## Data Availability

Data are contained within the article.

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
