# Peer review of "Antibacterial Potential of Microwave-Assisted Extraction Prepared Hydrolates from Different Salvia Species"

_plants, 2023, doi:10.3390/plants12061325_

Round 1

Reviewer 1 Report

Line 4: It is carnosic acid, speak about diterpenes with abietane skeleton, put after triterpenes as oleanic acid. organize the types of compounds found in the essentialk oil depend chemical structure.

Did not detect diterpenes on the analysis of  hydrolates? It had been reported as constituents of Salvia essential oils and also hydrolates: Plants (Basel). 2022 Jan 27;11(3):349. doi: 10.3390/plants11030349. 

Be careful with table 2, put all in on page. Explain better MIC50 and MIC90m meaning how calculated as footnote

The procedure to prepare the hydrolates, you did not concentrated then did no? Perhaps you could increase activity, see doi: 10.3390/molecules25235654

In conclusion, 264 line Salvia in italics. 

Author Response

- Line 44: It is carnosic acid, speak about diterpenes with abietane skeleton, put after triterpenes as oleanic acid. organize the types of compounds found in the essential oil depend chemical structure.

Accepted, will be adapted in the revised version of manuscript.

- Did not detect diterpenes on the analysis of  hydrolates? It had been reported as constituents of Salvia essential oils and also hydrolates: Plants (Basel). 2022 Jan 27;11(3):349. doi: 10.3390/plants11030349. 

In our work we focused mainly on monoterpenes in hydrolates of S. officinalis (92.79 %), S. nemorosa (87.7 %) and S. Aethiopis (91,86 %) . In hydrolates of S. sclarea and S. divinorum monoterpenes were in value 67.51 % and 72.48 %, and sesquiterpenes 15.11 % and 14.06 %. Further compounds were not analysed.

- Be careful with table 2, put all in on page.

Accepted, will be done in revised version.

- Explain better MIC50 and MIC90m meaning how calculated as footnote

In microbiology, the minimum inhibitory concentration (MIC) is the lowest concentration of a chemical, usually a drug, which prevents visible growth of a bacterium or bacteria. MIC depends on the microorganism. Similarly the lowest concentration of antimicrobial agent, that prevented minimal 90 % of bacterial growth compared with the growth in broth (medium without antimicrobial agent) was defined as MIC90. Similar concentration of antimicrobial agent that prevented minimal 50 % of bacterial growth compared with the growth in broth (medium without antimicrobial agent) was defined as MIC50. These both other output parameters (MIC90, MIC50) complete MIC parameter and contribute to better describing of the antibacterial potential of agents.

- The procedure to prepare the hydrolates, you did not concentrated then did no? Perhaps you could increase activity, see doi: 10.3390/molecules25235654

In time, dedicated to this study, we tested only crude hydrolates.

- In conclusion, 264 line Salvia in italics. 

Accepted.

Reviewer 2 Report

The submission by Ürgeová et al. reports on the antibacterial potential of some Salvia hydrolates. The work is routine, straightforward, and unexceptional but hydrolates do need to be studied on a par with essential oils, and in this respect they are lacking. So I suppose I can recommend acceptance subject to the following comments which the authors should pay close attention to.

1. The abstract is scattered in its description, e.g. S. nemorosa is not mentioned. Make it more methodical.

2. Line 17 (also line 159), S. divinorum was claimed to have no antibacterial effect, well, very mild in truth. Where to draw the line ? Seems arbitrary where the authors have decided it to be, is 326 that different to 217 ?

3. Authors are very careless with maintaining consistency of numbers: Line 13, should be 5.6 not 5.8, and 4.7 not 6.3 according to Table 1; line 164, should be 56.65.

4. Authors claimed 20 compounds were identified (line 79) altogether, no, only 13 were. And only 11 compounds were identified in S. divinorum, not 16 as stated.

5. Use either eucalyptol or 1,8-cineole as a name, not both.

6. Provide structures for the compounds.

7. Line 99, “We determine 100% bacterial growth in pure MHB and statistically compare this value with bacterial growth when treated with 100 antimicrobials, the hydrolates”, how ? Nonsense, nothing done.

8. Why are there on three growth inhibition plots ? Why not all 5 ?

9. For the antioxidant activity, why was a positive control not used ? This is a serious flaw.

10. Line 138, “More 138 than 10% of eucalyptol contains hydrolates of S. officinalis, S. nemorosa, and S. aethiopis.” Make it a proper statement, like the statement that immediately follows it.

11. Line 174, “at the lowest concentrations below 40%”, not true looking at the figures. Delete this nonsense.

12. Line 195, “hydrolates”.

13. Can delete section 4.7 as the only statistics done was to calculate average and standard deviation.

14. Line 272, just Gram positive ?

15. Line 141, also S. sclarea has thujone in greater than 10%.

Author Response

  1. The abstract is scattered in its description, e.g. nemorosa is not mentioned. Make it more methodical.

Corrected:

Salvia is a widely used herb that also contains essential oils and other valuable compounds. In this work, the five organic hydrolates of Salvia sp. were evaluated for their potential of antioxidant activity as well as antimicrobial activity against four bacterial strains. The hydrolates were obtained from fresh leaves by microwave-assisted extraction. Chemical composition analysis by GC-MS revealed that their major constituents were isopulegol (38.2 - 57.1%), further 1,8-cineole (6.3 - 19.6%), and thujone (5.8 - 14.1%). The minimum inhibitory concentration (MIC) of the plant hydrolates was tested by the microdilution method at concentrations ranging from 1.0 to 512 μg/ml. The hydrolates prepared from Salvia officinalis and Salvia sclarea showed inhibitory activity on the tested Gram-positive and Gram-negative bacteria, taxon Salvia nemorosa only partially. The hydrolate of Salvia divinorum had practically no antibacterial effect. Enterobacter asburiae was the only bacterium for which we found sensitivity to the hydrolate of Salvia aethiopis, with a MIC50 value of 216.59 µL/mL. The antioxidant activity of the hydrolates was low, ranging from 6.4 to 23.3%. Therefore, Salvia hydrolates could be used as antimicrobial agents in medicine, cosmetics and food preservation.

  1. Line 17 (also line 159), S. divinorum was claimed to have no antibacterial effect, well, very mild in truth. Where to draw the line ? Seems arbitrary where the authors have decided it to be, is 326 that different to 217 ?

Line 17 was corrected

In line 159 we described: We found no antimicrobial effect of the hydrolate of S. divinorum in this study, MIC50 and MIC90 were more than 500 µL/mL for three bacteria - E. coli, M. luteus and B. subtilis, and a very low antimicrobial effect on E. asburiae with a MIC50 value of 325.61 µL/mL. Similarly, this strain was sensitive to S. aethiopis hydrolate, with a MIC50 value of 216.59 µL/mL.

  1. Authors are very careless with maintaining consistency of numbers: Line 13, should be 5.6 not 5.8, and 4.7 not 6.3 according to Table 1; line 164, should be 56.65.

Accepted, will be corrected.

  1. Authors claimed 20 compounds were identified (line 82) altogether, no, only 13 were. And only 11 compounds were identified in S. divinorum, not 16 as stated.

Will be corrected as follows:  In total, from 20 different possible compounds were identified 13 compounds. Most compounds, eleven, were identified in the hydrolates of S. divinorum and S. sclarea, and at least six in the hydrolate of S. officinalis.

  1. Use either eucalyptol or 1,8-cineole as a name, not both.

Accepted, will be corrected.

  1. Provide structures for the compounds.

Will be supplemented for main compounds – isopulegone; 1,8-cineole and thujone.

  1. Line 99, “We determine 100% bacterial growth in pure MHB and statistically compare this value with bacterial growth when treated with 100 antimicrobials, the hydrolates”, how ? Nonsense, nothing done.

Statistical analysis was performed by probit analysis, based on the measured absorbance on microplates; we determined the percentage of bacterial growth in hydrolates presence.

  1. Why are there on three growth inhibition plots ? Why not all 5 ?

In the work, based on measurement of MIC, we graphically illustrated the comparison of the effects of hydrolates, which acted on at least two bacterial strains. The hydrolates of S. divinorum and S. aethiopis affect only on E. aesburie, what we mention on the line 169 and 170.

  1. For the antioxidant activity, why was a positive control not used ? This is a serious flaw.

It has been applied TROLOX (6-Hydroxy-2,5,7,8-tetramethylchroman-2-carboxylic acid) as a standard for assay of antioxidant activity and expressed as TEAC value. The data will be completed as follows:

Tested hydrolates

Antioxidant activity (%)

TEAC (mg/ml)

S. officinalis

23.25 ± 0.26

52,16 ± 2,85

S. nemorosa

14.61 ± 0.09

31,08 ± 1.09

S. aethiopis

10.62 ± 0.96

24,12 ± 0.96

S. sclarea

18.99 ± 0.58

42,12 ± 1.58

S. divinorum

6.42 ± 0.57

14,41 ± 0.57

  1. Line 138, “More 138 than 10% of eucalyptol contains hydrolates of S. officinalis, S. nemorosa, and S. aethiopis.” Make it a proper statement, like the statement that immediately follows it.

It was supplemented with data on the amount of 1,8-cineole.

  1. Line 174, “at the lowest concentrations below 40%”, not true looking at the figures. Delete this nonsense.

Will be corrected (line 208) as follows: The hydrolates of S. officinalis and S. nemorosa were able, at the lowest concentrations, partially inhibit the growth of Gram-positive strains throughout the cultivation.

  1. Line 195, “hydrolates”.

Accepted, will be corrected

  1. Can delete section 4.7 as the only statistics done was to calculate average and standard deviation.

Statistical analysis was performed by probit analysis, based on the measured absorbance on microplates; we determined the percentage of bacterial growth in hydrolates presence.

  1. Line 272, just Gram positive ?

Will be corrected (line 316) as follows: The interesting antimicrobial activity of S. sclarea and S. nemorosa hydrolates is also described.

  1. Line 141, also S. sclarea has thujone in greater than 10%.

Accepted, will be corrected

Reviewer 3 Report

General comments:

The material and methods regarding the extraction process and the chromatography are not described sufficiently. It is not clear how these were carried out.

The presentation of the results could be improved.

This study could be more informative if the essential oils of the five Salvia species were also examined for comparison.

It is not clear which of the compounds identified in the hydrolates are responsible for the inhibition of bacterial growth.

The presence of isopulegol in the Salvia hydrolates does not appear to have been reported in other research papers and since it is the main compound in all of the hydrolates it cannot be responsible for the antibacterial activity.

Some points are covered more specifically below:

Title

Give abbreviation MAE in full in title.

Use species not sp. in title. “sp.” is not an abbreviation for the word species. It means not identified at species level.

Abstract

The abstract covers the study.

What does “organic hydrolate” mean exactly?

Line 9. “Salvia sp.” Use “of five Salvia species”. “Salvia sp.” Means not identified at species level. This also occurs in several other places in the text.

Keywords:

Introduction

The introduction is generally comprehensive of the topic.

Outline what are the hydrosols which are mentioned in the references.

Materials and Methods

The Materials and Methods should be improved. Some comments below:

There are insufficient details about the microwave extraction process. What equipment was used and how it was carried out.

There are insufficient details about the gas chromatography. The type of column used. Column conditions and temperature programme used, and equipment used.

How were the RI calculated? There is no indication for alkane standards.

Results

Table 1.

Use capital S for Salvia in title

It is not necessary to use the symbol % after each number in the Table put this in the column header.

Use “unknown” or “unidentified” or “mass/charge ratio” for compounds indicated by hyphens/dashes.

Use “Total” in last row

Was the compound at RI 1117 a- or b- thujone?

The amount of the compound at RI 1146 seems very high. Was it really iso-pulegol? The minus sign is also bigger here than in other compounds

Were RI calculated using n-alkane standards?

Figure 1, 2, 3. The bacteria names are not given in italics.

Why are the concentrations not presented from 0 to 500 ie in increasing concentration? This is rather confusing for the reader. Explain what is bacterial growth (%).

Why is the 120% line given in Fig 3. Is the maximum possible not 100%.

Statistics: Values are means +/- sd?

Discussion

Line 125. Use Salvia not Salvia sp. And elsewhere.

The literature provided in the discussion does not appear to support the presence of isopulegol in the hydrolates of other Salvia species. Nor is this discussed that the main constituent in the hydrolates of the five Salvias was isopulegol. It also cannot be assumed to be responsible for the antibacterial activity as there were differential inhibitions. Please comment.

Line 155. “Antibacterial” activity

Authors should conclude whether the aims of the research have been achieved.

Conclusions

Authors should conclude whether the aims of the research have been achieved.

References

Species names should be in italics in all titles, see ref 4, 7, 9 and others.

Author Response

Title

Give abbreviation MAE in full in title.

Use species not sp. in title. “sp.” is not an abbreviation for the word species. It means not identified at species level.

Accepted, title will be corrected as follows:

Antibacterial potential of via microwave-assisted extraction prepared hydrolates from different Salvia species.

 Abstract

The abstract covers the study.

What does “organic hydrolate” mean exactly?

The word organic wil be eliminated.

Line 9. “Salvia sp.” Use “of five Salvia species”. “Salvia sp.” Means not identified at species level. This also occurs in several other places in the text.

Accepted, will be corrected

Introduction

The introduction is generally comprehensive of the topic.

Outline what are the hydrosols which are mentioned in the references.

We wrote on line 62: A hydrolate (H) is internationally defined as the distilled aromatic water that remains after hydro- or steam distillation and separation of the essential oil.

Materials and Methods

The Materials and Methods should be improved. Some comments below:

There are insufficient details about the microwave extraction process. What equipment was used and how it was carried out.

Will be corrected as follows:

The extraction was carried out in a microwave oven Bosch FFL023MS2 (Gerlingen, Germany) at 800 W for 8 minutes in EssenEx® 100A Essential Oil Extraction Kit (Oregon, USA). We used ice to condense the steam and oil fraction.

There are insufficient details about the gas chromatography. The type of column used. Column conditions and temperature programme used, and equipment used.

How were the RI calculated? There is no indication for alkane standards.

Will be corrected as follows:

The analytes were identified and the relative composition of the Salvia hydrolates was determined by gas chromatography followed by mass spectrometry (GC/MS) as described by [53]. Prior to injection, hydrolates were extracted in hexane in ratio 1.5:1 and dried over anhydrous sodium sulphate according to [38]. Analyses were carried out using an Agilent 7890A GC coupled to an Agilent  MSD5975C  MS  detector  (Agilent  Technologies,  Palo  Alto,  CA,  USA) with  a  HP-5MS  column  (30  m  ×  0.25  mm,  0.25  mm  film thickness). The analytical conditions were as follows: injector temperature 250°C; the oven temperature from 60°C to 231°C at 3°C/min. Analytes were identified by matching their mass spectra and retention indices with those of authentic samples and/or NIST /Wiley spectra libraries and available literature data [54]. Relative proportions were calculated by dividing the individual peak area by the total area of all peaks. Only compounds with more than 1% were considered.

Results

Table 1.

Use capital S for Salvia in title

Accepted, will be corrected

It is not necessary to use the symbol % after each number in the Table put this in the column header.

We accept your suggestion; but for formatting of the table it is favorable used the symbol % after values.

Use “unknown” or “unidentified” or “mass/charge ratio” for compounds indicated by hyphens/dashes.

Accepted, will be corrected

Use “Total” in last row

Accepted, will be corrected

Was the compound at RI 1117 a- or b- thujone?

Will be corrected as follows:

Sum of both epimers, α- and β-Thujone

The amount of the compound at RI 1146 seems very high. Was it really iso-pulegol? The minus sign is also bigger here than in other compounds

Yes we verified GC analysis again, it is really iso-pulegol, the minus sign will be corrected

Were RI calculated using n-alkane standards?

Analytes were identified by matching their mass spectra and retention indices with those of authentic samples and/or NIST /Wiley spectra libraries and available literature data. Compounds with index a were identified and confirmed by co-injection of the authentic standard.

Figure 1, 2, 3. The bacteria names are not given in italics.

Accepted, will be corrected

Why are the concentrations not presented from 0 to 500 ie in increasing concentration? This is rather confusing for the reader. Explain what is bacterial growth (%).

With decreasing concentration decrease antimicrobial effect of hydrolates and the density/growth of bacteria increases. The bacterial growth it is yet explained within the response to other reviewer.

Why is the 120% line given in Fig 3. Is the maximum possible not 100%.

The Y axis value 120% it is really only virtual, it is given into this chart for better demonstration and presentation of line near value 100%. Hypothetically, some samples could be promoters of the growth; hypothetically, some values could lie over 100%, but not in this case.

Statistics: Values are means +/- sd?

Yes, each value is given as mean ± standard deviation.

Discussion

Line 125. Use Salvia not Salvia sp. And elsewhere.

Accepted, will be corrected.

The literature provided in the discussion does not appear to support the presence of isopulegol in the hydrolates of other Salvia species. Nor is this discussed that the main constituent in the hydrolates of the five Salvias was isopulegol. It also cannot be assumed to be responsible for the antibacterial activity as there were differential inhibitions. Please comment.

Isopulegol is found in the essential oils of a variety of fruits and herbs including orange, lemongrass, lemon balm, canabis, mint, geranium, and eucalyptus. Isopulegol acts as a pest deterrent for plants that produce it. While keeping the plant safe from damage, the compound also attracts pollinators to ensure the plant’s survival and ability to thrive in the wild. We can assume that Salvia may also contain isopulegol.

Further studies are necessary to discover the extent of isopulegol’s health benefits, but research has shown promising links between isopulegol and anti-inflammatory [1], antiviral, antioxidant [2] and gastroprotective effects [3]. 

  1. Bounihi A, Hajjaj G, Alnamer R, Cherrah Y, Zellou A. In Vivo Potential Anti-Inflammatory Activity of Melissa officinalis L. Essential Oil. Adv Pharmacol Sci. 2013;2013:101759. doi: 10.1155/2013/101759. Epub 2013 Dec 5. PMID: 24381585; PMCID: PMC3870089.
  2. Silva MI, Silva MA, de Aquino Neto MR, Moura BA, de Sousa HL, de Lavor EP, de Vasconcelos PF, Macêdo DS, de Sousa DP, Vasconcelos SM, de Sousa FC. Effects of isopulegol on pentylenetetrazol-induced convulsions in mice: possible involvement of GABAergic system and antioxidant activity. Fitoterapia. 2009 Dec;80(8):506-13. doi: 10.1016/j.fitote.2009.06.011. Epub 2009 Jun 25. PMID: 19559770.
  3. Silva MI, Moura BA, Neto MR, Tomé Ada R, Rocha NF, de Carvalho AM, Macêdo DS, Vasconcelos SM, de Sousa DP, Viana GS, de Sousa FC. Gastroprotective activity of isopulegol on experimentally induced gastric lesions in mice: investigation of possible mechanisms of action. Naunyn Schmiedebergs Arch Pharmacol. 2009 Sep;380(3):233-45. doi: 10.1007/s00210-009-0429-5. Epub 2009 May 29. PMID: 19479241.

Line 155. “Antibacterial” activity

Acepted, will be corrected.

Conclusions

Authors should conclude whether the aims of the research have been achieved.

Accepted, to Conclusion chapter will be add:

Our postulated aims of the research in this study were achieved, we wanted to remark differences between various Salvia varieties as from chemical composition, too from antibacterial and antioxidant activity point of view, what was proved and described.

References

Species names should be in italics in all titles, see ref 4, 7, 9 and others.

Accepted, will be corrected.

Reviewer 4 Report

Why 2 species of Salvia were not tested for antimicrobial activity?

Where are the results from  of minimum inhibitory concentration?

The work needs improvement with other determinations, for example determination of flavonoids, total polyphenols, etc.

Author Response

- Why 2 species of Salvia were not tested for antimicrobial activity?

All species of Salvia were tested for antimicrobial activity (Table 2). In the work, based on measurement of MIC, we graphically illustrated the comparison of the effects of hydrolates, which acted on at least two bacterial strains. The hydrolates of S. divinorum and S. aethiopis affect only on E. aesburie, what we mention on the line 169 and 170.

- Where are the results from  of minimum inhibitory concentration?

The results of MIC are in the Table 2.

- The work needs improvement with other determinations, for example determination of flavonoids, total polyphenols, etc.

We tested samples of hydrolates for comparing them with essential oils. The subject of this our study was different from mentioned categories of secondary metabolites, in other papers we applied the determination of total polyphenols/phenolic acid/flavonoids/amine/thiols content.

Round 2

Reviewer 2 Report

The authors have made a sufficient number of improvements to the manuscript.

Reviewer 3 Report

1)      Line 62 Include “hydrolates, also known as hydrosols,”

2)      It is suggested that text given below in answer to a reviewer’s comment should be incorporated into the text together with the references. The explanation should not have been intended only for the reviewer.

“Isopulegol is found in the essential oils of a variety of fruits and herbs including orange, lemongrass, lemon balm, canabis, mint, geranium, and eucalyptus. Isopulegol acts as a pest deterrent for plants that produce it. While keeping the plant safe from damage, the compound also attracts pollinators to ensure the plant’s survival and ability to thrive in the wild. We can assume that Salvia may also contain isopulegol.

Further studies are necessary to discover the extent of isopulegol’s health benefits, but research has shown promising links between isopulegol and anti-inflammatory [1], antiviral, antioxidant [2] and gastroprotective effects [3]. 

  1. Bounihi A, Hajjaj G, Alnamer R, Cherrah Y, Zellou A. In Vivo Potential Anti-Inflammatory Activity of Melissa officinalis L. Essential Oil. Adv Pharmacol Sci. 2013;2013:101759. doi: 10.1155/2013/101759. Epub 2013 Dec 5. PMID: 24381585; PMCID: PMC3870089.
  2. Silva MI, Silva MA, de Aquino Neto MR, Moura BA, de Sousa HL, de Lavor EP, de Vasconcelos PF, Macêdo DS, de Sousa DP, Vasconcelos SM, de Sousa FC. Effects of isopulegol on pentylenetetrazol-induced convulsions in mice: possible involvement of GABAergic system and antioxidant activity. Fitoterapia. 2009 Dec;80(8):506-13. doi: 10.1016/j.fitote.2009.06.011. Epub 2009 Jun 25. PMID: 19559770.
  3. Silva MI, Moura BA, Neto MR, Tomé Ada R, Rocha NF, de Carvalho AM, Macêdo DS, Vasconcelos SM, de Sousa DP, Viana GS, de Sousa FC. Gastroprotective activity of isopulegol on experimentally induced gastric lesions in mice: investigation of possible mechanisms of action. Naunyn Schmiedebergs Arch Pharmacol. 2009 Sep;380(3):233-45. doi: 10.1007/s00210-009-0429-5. Epub 2009 May 29. PMID: 19479241.”

3)      Table 3 a column with data has been added. Are the method and results described in the text?

Give in full the abbreviation TEAC

Reviewer 4 Report

The work could be improved with other analyses